# Morphology and Phylogeny Reveal Three *Montagnula* Species from China and Thailand

**DOI:** 10.3390/plants12040738

**Published:** 2023-02-07

**Authors:** Ya-Ru Sun, Jing-Yi Zhang, Kevin D. Hyde, Yong Wang, Ruvishika S. Jayawardena

**Affiliations:** 1School of Science, Mae Fah Luang University, Chiang Rai 57100, Thailand; 2Department of Plant Pathology, College of Agriculture, Guizhou University, Guiyang 550025, China; 3Center of Excellence in Fungal Research, Mae Fah Luang University, Chiang Rai 57100, Thailand; 4Innovative Institute for Plant Health, Zhongkai University of Agriculture and Engineering, Guangzhou 510225, China

**Keywords:** two new species, multi-gene phylogeny, Pleosporales, sexual morph, taxonomy

## Abstract

Four stains were isolated from two fresh twigs of *Helwingia himalaica* and two dead woods during investigations of micro-fungi in China and Thailand. Phylogenetic analyses of four gene regions LSU, ITS, SSU and tef1-*α* revealed the placement of these species in *Montagnula*. Based on the morphological examination and molecular data, two new species, *M. aquatica* and *M. guiyangensis*, and a known species *M. donacina* are described. Descriptions and illustrations of the new collections and a key to the *Montagnula* species are provided. *Montagnula chromolaenicola*, *M. puerensis*, *M. saikhuensis*, and *M. thailandica* are discussed and synonymized under *M. donacina*.

## 1. Introduction

Didymosphaeriaceae (Pleosporales) was established by Munk [1] and with Didymosphaeria as the type genus. There are 33 genera accepted in this family based on morphology and phylogenetic analyses [2,3]. Species belonging to Didymosphaeriaceae have a wide geographical and host distribution and have different modes of nutrition, such as saprobic on plant litter, herbaceous stems, or in soil; endophytic on healthy leaves or twigs; and pathogenic on plants, animals, or humans [2,4,5,6,7,8,9].

Berlese [10] introduced *Montagnula*, typified by *M. infernalis*, which has bitunicate asci and dictyosporous ascospores. Around a century later, Crivelli [11] refined *Pleospora* and transferred eight *Pleospora* species and one *Teichospora* species to *Montagnula* based on morphology. Leuchtmann [12] included phragmosporous and didymosporous species in this genus, making species identification heterogeneous. Aptroot [13] established *Munkovalsaria* to accommodate *Mu. donacina* based on valsoid ascomata, bitunicate, fissitunicate asci, and 1-septate ascospores, however, Wanasinghe et al. [14] synonymized *Munkovalsaria* under *Montagnula* based on analyses of combined LSU, SSU, and ITS sequence data. Crous et al. [7] reported the first coelomycetous asexual morph species *M. cylindrospora* in this genus. So far, there are 39 validly published *Montagnula* species in Species Fungorum (accessed on 28 January 2023) [15]. However, only 18 species have molecular data. Morphologically, sexual morphs of *Montagnula* have three different types of ascospores (didymospore, phragmospore, and dictyospore) [8,16]. Phylogenetically, species with the same type of ascospore tend to cluster together [9,17]. In recent years, there have been many reports on *Montagnula* species [8,9,18,19,20], but there are very few comprehensive and systematic papers.

*Montagnula* species occur on terrestrial habitats with a wide geographic and host distribution [8,21]. Most *Montagnula* species have been found on dead leaves and twigs by their sexual morph [8,10,17,18,21,22,23]. The sexual morph is characterized by globose to pyriform, immersed to erumpent or superficial, brown to dark brown ascomata with or without ostiole, *textura angularis* peridium. Asci are cylindric-clavate to clavate, bitunicate, and 2–8-spored, and ascospores are pale to dark brown, phragmosporous, didymosporous, or dictyosporous [8,10,16,19,20]. Only one species has been reported as a coelomycetous asexual morph, which has solitary, superficial, brown to dark brown, globose to subglobose conidiomata, phialidic, ampulliform to dolioform, hyaline conidiogenous cells, and aseptate, hyaline, cylindrical conidia [7].

To study the taxonomy and diversity of *Montagnula* species, four *Montagnula* specimens were obtained from terrestrial and freshwater habitats in China and Thailand. Based on the morphological examination and phylogenetic analyses, two new species, viz. *M. aquatica* and *M. guiyangensis*, and a known species, *M. donacina* are introduced with illustrations and descriptions. We also provide a key to *Montagnula* species.

## 2. Results

### 2.1. Phylogenetic Analyses

Phylogenetic relationships of four *Montagnula* species were evaluated in the multi-gene analysis of 59 Didymosphaeriaceae strains. Two strains of *Fuscostagonospora* (Fuscostagonosporaceae), *F. sasae* (HHUF 29106) and *F. cytisi* (MFLUCC 16–0622), were selected as the outgroup taxa. The analyzed alignment consisted of combined LSU (1–801 bp), ITS (802–1301 bp), SSU (1302–2287 bp), and tef1-α (2288–3127) sequence data, including gaps. The most likely tree (−ln = 17,057.307078) is presented (Figure 1) to show the phylogenetic placements of the new taxa.

The ML and BYPP trees (not shown) were similar in topology. The genus *Montagnula* formed an independent topmost clade in the phylogenetic tree. *Montagnula* species were divided into four clades in the phylogenetic tree. Our four strains nested within the genus and represented three species. *Montagnula aquatica* (MFLU 22–0171) was placed in Clade 2. Two *M. guiyangensis* strains (HKAS 124556 and HGUP 22–0800) clustered together with ML-BS = 100%, BYPP = 1.00 support and formed a distinct lineage in Clade 3. Our isolate HKAS 124552 clustered together with *M. donacina* in Clade 1.

### 2.2. Taxonomy

*Montagnula aquatica* Y.R. Sun, Yong Wang bis and K.D. Hyde, sp. nov. Figure 2.Index Fungorum number: IF900129; Facesoffungi number: FoF 12922.Holotype: MFLU 22−0171.Etymology: Referring to the aquatic habitat of the fungus.

Saprobic on submerged decaying wood in freshwater habitat. Sexual morph: *Ascomata* 250–430 μm long, 250–340 μm high, semi-immersed, solitary or scattered, globose, uniloculate, black, smooth-walled, with a central ostiole. *Ostiole* papillate, central. *Peridium* 10–22 μm wide, fused with host tissues, comprising two layers of pale brown to brown cells of textura angularis. *Hamathecium* comprising 1–2 μm wide, numerous filamentous, branched, hyaline, septate, guttulate, pseudoparaphyses. *Asci* 110–130 × 13–19 μm ($\bar{x}$ = 122 × 15.5 μm, n = 10), bitunicate, 8-spored, cylindric-clavate, slightly curved, short-stalked. *Ascospores* 24–35 × 7.5–14 μm ($\bar{x}$ = 30.5 × 10.5 μm, n = 30), hyaline to yellow-brown when immature, dark brown when mature, 2-seriate, fusiform to broadly fusiform, 3-septate, widest at the center, tapering towards ends, conical both ends, guttulate, without appendages and mucilaginous sheath. Asexual morph: Not observed.

Culture characteristics: Ascospores germinated on PDA within 12 h at 25 °C. Germ tubes produced from both ends. Colonies on PDA reached 5 cm diam. after 3 weeks at 25 °C; mycelium white, flossy, circular, with the entire edge; white to yellow in reverse.

Material examined: Thailand, Chiang Rai Province, Bandu District, saprobic on decaying wood submerged in a river in an unknown waterfall, 6 March 2021, Y.R. Sun, 26 (MFLU 22–0171, holotype).

Notes: Morphologically, *M. aquatica* can be distinguished by its larger ascospores from its related species in Clade 3 (Figure 1) (24–35 × 7.5–14 μm in *M. aquatica* vs. 18–25 × 5–88 μm in *M. camporesii* vs. 18–22.5 × 6.5–9.5 μm in *M. cirsii* vs. 20–23 × 7–9 μm in *M. scabiosae*) [19,24,25]. In addition, *M. aquatica* has thinner peridia than *M. cirsii* (10–22 μm vs. 41–58.5 μm) and has larger asci than *M. camporesii* (110–130 × 13–19 μm vs. 80–120 × 10–15 μm) [19,25]. The results of base pair differences (Table 1) also support the establishment of *M. aquatica* as a new species [26,27]. Thus, *M. aquatica* sp. nov is introduced and it is the first *Montagnula* species reported from freshwater habitats.

*Montagnula guiyangensis* Y.R. Sun, Yong Wang bis and K.D. Hyde, sp. nov. Figure 3.Index Fungorum number: IF900130; Facesoffungi number: FoF 12923.Holotype: HKAS 124556.Etymology: Referring to the location in which the fungus was collected.

Saprobic on twigs of *Helwingia himalaica* in terrestrial habitat. Sexual morph: *Ascomata* 300–400 × 350–400 μm, semi-immersed, solitary or scattered, globose, uniloculate, black, with a central ostiole. *Ostiole* papillate, central. *Peridium* 20–40 μm wide, fused with host tissues, comprising two layers of pale brown to brown cells of textura angularis. *Hamathecium* comprises 1.5–3 μm wide, branched, hyaline, septate, pseudoparaphyses. *Asci* 84–135 × 10–15 μm ($\bar{x}\;$= 104 × 12 μm, n = 15), bitunicate, 8-spored, clavate, with a short, bulbous long pedicel, slightly curved. *Ascospores* 10–20 × 3.5–6 μm ($\bar{x}$ = 15.5 × 5 μm, n = 35), hyaline to olivaceous when immature, brown when mature, overlapping uniseriate or 2-seriate, fusiform, 1-septate, constricted at the septum, slightly widest at the upper cell and tapering towards ends, guttulate, sheath drawn out to form polar appendages, from both ends of the ascospores, straight or slightly curved. Asexual morph: Not observed.

Culture characteristics: Ascospores germinated on PDA within 12 h at 25 °C. Germ tubes produced from both ends. Colonies on PDA reached 7 cm diam after four weeks at 25 °C, mycelium white to gray, flossy, circular, undulate, yellow in reverse.

Material examined: China, Guizhou Province, Guiyang City, Nanming District, Guiyang Medicinal Botanical Garden, on twigs of *Helwingia himalaica*, 22 December 2021, Y.R. Sun, 22-41 (HKAS 124556, holotype; ex-type living culture GUCC 816); ibid, on twigs of *Helwingia himalaica*, 22 December 2021, Y.R. Sun, 41-2 (HGUP 22–800, paratype; living culture GUCC 22–0817).

Notes: *Montagnula guiyangensis* was isolated from *Helwingia himalaica*, an important medicinal plant. Multi-gene analyses showed that *M. guiyangensis* is a phylogenetically distinct species in Clade 3 (Figure 1). Morphologically, *M. guiyangensis* resembles *M. appendiculata*, *M. chiangraiensis*, and *M. chromolaenae* in having fusiform, 1-septate ascospores with appendages. *Montagnula guiyangensis*, however, differs by its larger ascomata from *M. chromolaenae* and *M. appendiculata* (300–400 × 350–400 μm in *M. guiyangensis* vs. 170–190 × 170–190 μm in *M. chromolaenae* vs. 100–200 μm in *M. appendiculata*) [8]. *Montagnula guiyangensis* has larger asci than *M. chiangraiensis* (84–135 × 10–15 μm vs. 60–75 × 8–11 μm). *Montagnula guiyangensis* can be distinguished from *M. aloes* by 1-septate, fusiform ascospores with appendages, while the latter has 3-septate, ovoid to ellipsoid ascospores [22]. In addition, comparisons of ITS, LSU, and SSU sequences between *M. guiyangensis* and phylogenetically related species are provided in Table 2 (tef1-*α* not available for *M. aloes*, *M. appendiculata*, *M. chiangraiensis,* and *M. chromolaenae*).

*Montagnula donacina* (Niessl) Wanas., E.B.G. Jones and K.D. Hyde, Fungal Biology 120 (11): 1365 (2016) Figure 4.=*Montagnula chromolaenicola* Mapook and K.D. Hyde.=*Montagnula puerensis* Tibpromma and Du.=*Montagnula saikhuensis* Wanas., E.B.G. Jones and K.D. Hyde.=*Montagnula thailandica* Mapook and K.D. Hyde.Index Fungorum number: IF557299; Facesoffungi number: FoF 07792.

Saprobic on decaying wood in terrestrial habitat. Sexual morph: *Ascomata* 405–470 μm high, 280–380 μm wide, semi-immersed, solitary or scattered, globose, uniloculate, black, with a central ostiole. *Ostiole* papillate, central. *Peridium* 15–30 μm wide, fused with host tissues, comprising of two layers of pale to brown cells of textura angularis. *Hamathecium* comprising 1–2.5 μm wide, branched, hyaline, septate, pseudoparaphyses. *Asci* 80–125 × 9–12 μm, bitunicate, 8-spored, clavate, with a bulbous long pedicel, slightly curved. *Ascospores* 10–15 × 4–7 μm ($\bar{x}$ = 13.5 × 5.5 μm, n = 30), brown, overlapping uniseriate or 2-seriate, fusiform, 1-septate, constricted at the septum, slightly widest at the upper cell and tapering towards ends, guttulate, straight or slightly curved. Asexual morph: Not observed.

Culture characteristics: Ascospores germinated on PDA within 12 h at 25 °C. Germ tubes produced from one side of the middle of ascospore. Colonies on PDA reached 5 cm diam after four weeks at 25 °C, mycelium white to gray, flossy, circular, undulate, gray in reverse.

Material examined: China, Guizhou Province, Qianxinan Bouyei and Miao Autonomous Prefecture, Anlong County, on dead wood, 16 March 2022, J.Y. Zhang, Y312 (HKAS 124552; living culture GUCC 22–0818).

Notes: *Montagnula chromolaenicola*, *M. donacina*, *M. puerensis*, *M. saikhuensisi,* and *M. thailandica* clustered together without obvious branches in the phylogenetic tree (Figure 1). Morphologically, they have similar ascomata, asci, and ascospores, including measurement size (Table 3). It is worth noting that Wanasinghe et al. [14] took multi-loculate ascomata as the difference between *M. donacina* and *M. saikhuensis*. Du et al. [21] distinguished *M. donacina* and *M. puerensis* by *M. donacina* having carbonaceous ascostromata. However, the previous literature did not mention that *M. donacina* has multi-loculate, carbonaceous ascostromata [13,28]. Comparisons of ITS, LSU, SSU, and tef1-*α* sequences between *M. donacina* and phylogenetically related species are provided in Table 4. Few differences exist among their ITS, LSU, and SSU sequences, respectively, and there is a maximum difference of 10 bp in tef1-α gene. We conclude that the evidence for these five species as independent species is insufficient. The slight difference of multi-genes may represent the intraspecific variation. Therefore, we synonymize *M. chromolaenicola*, *M. puerensis*, *M. saikhuensisi*, and *M. thailandica* under *M. donacina* based on the nomenclatural priority. Our new collection HKAS 124552 has overlapping characteristics with these *M. donacina* isolates. Phylogenetically, HKAS 124552 grouped with them in Clade 1 (Figure 1). Thus, we identify our isolate as *M. donacina*.

## 3. Discussion

*Montagnula* species have a worldwide distribution that has been reported from America, Australia, Bahamas, China, Italy, Portugal, and Thailand [8,21]. Previous literature reported that all *Montagnula* species have been derived from terrestrial habitats [7,8,10,11,20,21,23]. We introduced a freshwater *Montagnula* species here that broke the record of the monolithic habitat for *Montagnula* species. These species have various hosts, such as *Agave* sp., *Pandanus* sp., and *Ilex* sp. [5,21]. However, rarely have studies focused on fungi associated with *H. himalaica* (Helwingiaceae). *Helwingia himalaica* is distributed in Bhutan, China, Nepal, and Thailand (https://www.havlis.cz/karta_en.php?kytkaid=5087 accessed on 27 January 2023). It has a high medicinal value that is used to treat colds, coughs, stomach pains, and fractures. In this study, we introduced a new species, *M. guiyangensis*, which was isolated from *H. himalaica*.

*Montagnula* species had didymosporous, phragmosporous, and dictyosporous ascospores [8,14,25]. Species with the same type of spores tended to cluster together (Figure 1). In our phylogenetic study, *Montagnula* species were divided into four major phylogenetic clades. (Figure 1). Four didymosporous species (*M. acaciae, M. donacina*, *M. graminicola*, and *M. opulenta*) and a coelomycetous asexual morph species, *M. cylindrospora*, were placed in Clade 1. *Montagnula acaciae, M. donacina*, and *M. opulenta* had didymospores without sheath but *M. graminicola* was surrounded by a sheath. Species in Clade 2 generally had fusiform to broadly fusiform phragmospores. Although the morphological characteristics of *M. jonesii* matched well with the species in Clade 2, it formed a distinct and basal clade in the tree. Species in Clade 3 had didymospores with polar appendages or were surrounded by a sheath, except for *M. aloes,* which had phragmospores without appendages. However, it is worth noting that the characteristics of *M. aloes* were observed from the culture, whereas other species were observed from the natural substrates. Fresh collections of *M. aloes* from nature are necessary to resolve the issue. However, there are no sequences available for dictyosporous species, e.g., *M. dasylirionis*, *M. mohavensis*, and *M. yuccigena*. Therefore, whether these species would gather in one clade cannot be inferred. Future molecular studies, incorporating a broad sampling of *Montagnula* and other Didymosphaeriaceae species, may separate *Montagnula* into several new genera based on the septation of the ascospores.

## 4. Materials and Methods

### 4.1. Collection, Examination, and Isolation

The fresh samples were collected in China and Thailand from 2019 to 2022. Samples were brought to the laboratory in Ziplock plastic bags for examination, as described in Senanayake et al. [29]. The fruiting bodies on natural substrates were observed and photographed using a stereomicroscope (SteREO Discovery, V12, Carl Zeiss Microscopy GmBH, Berlin, Germany; VHX-7000, Keyence, Osaka, Japan). Morphological characters were observed using a Nikon ECLIPSE Ni compound microscope (Nikon, Tokyo, Japan) photographed with a Nikon DS-Ri2 digital camera (Nikon, Japan), and Carl Zeiss compound microscope (Carl Zeiss AG, Germany) photographed with an Axiocam 208 color digital camera (Carl Zeiss AG, Germany). The photo plates were made by the Adobe Photoshop CS6 Extended v. 13.0 software. Measurements were done with the Tarosoft (R) Image Frame Work Version 0.9.7 software.

Single spore isolation was used to obtain pure cultures following the methods described by Senanayake et al. [29]. Germinated ascospores were transferred to new potato dextrose agar (PDA) plates and incubated at 25°C for 4 weeks. The pure cultures obtained were deposited in Mae Fah Luang University Culture Collection (MFLUCC), Chiang Rai, Thailand, and the Guizhou University Culture Collection (GUCC), Guiyang, China. Herbaria materials were deposited in the herbarium of Mae Fah Luang University (MFLU), Chiang Rai, Thailand, and the Kunming Institute of Botany Academia Sinica (HKAS), Kunming, China. Facesoffungi (FoF) and Index Fungorum numbers were acquired as described in Jayasiri et al. [30] and Index Fungorum (2023) [31]. Records were added to the Mekong Subregion (GMS) database [32]. The establishment of new species was decided upon the recommendations of Chethana et al. [27] and Jayawardena et al. [33].

### 4.2. DNA Extraction, PCR Amplification, and Sequencing

PrepManTM Ultra Sample Preparation Reagent (Thermo Fisher Scientific, Yokohama, Japan) was used to extract DNA directly from fruiting bodies. BIOMIGA Fungus Genomic DNA Extraction Kit (Biomiga, San Diego, CA, USA) was used to extract DNA from fresh fungal mycelia, which were grown on PDA medium for 4 weeks at 25 °C. Three genes were selected in this study: the large subunit nuclear ribosomal DNA (LSU), the small subunit nuclear ribosomal DNA (SSU), the internal transcribed spacers (ITS), and the translation elongation factor 1 (tef1-α). Polymerase chain reaction (PCR) was carried out in 20 μL reaction volume, which contained 10 μL 2 × PCR Master Mix, 7 μL ddH_2_O, 1 μL of each primer, and 1 μL template DNA. The PCR thermal cycle program and primers are given in Table 5. Purification and sequencing of PCR products were carried out at SinoGenoMax (Beijing) Co., China.

### 4.3. Phylogenetic Analyses

BLASTn (https://blast.ncbi.nlm.nih.gov//Blast.cgi, accessed on 27 January 2023) was used to evaluate closely related strains to our new taxa. Other sequences used in this study were obtained from GenBank referring to Mapook et al. [8] and Du et al. [21] (Table 6). The single gene sequences were viewed using BioEdit v. 7.0.9.0 [37]. Alignments for each locus were generated with MAFFT v.7 (https://mafft.cbrc.jp/alignment/server/, accessed on 27 January 2023) and manually improved using AliView [38] for maximum alignment and minimum gaps. The final single-gene alignments were combined by SequenceMatrix 1.7.8 [39]. The single locus and combined analyses were carried out for maximum likelihood (ML) and Bayesian posterior probability (BYPP).

The ML analyses were performed in CIPRES [40] with RAxML-HPC v. 8.2.12 [41] using a GTRGAMMA approximation with rapid bootstrap (BS) analysis followed by 1000 bootstrap replicates.

The BYPP analyses were conducted in CIPRES [40] with MrBayes on XSEDE 3.2.7a [42]. The best nucleotide substitution model for each data partition was evaluated by MrModeltest 2.2 [43]. The substitution model GTR+I+G was decided for LSU, ITS, and SSU sequences. The Markov chain Monte Carlo (MCMC) sampling approach was used to calculate posterior probabilities (PP) [44]. Six simultaneous Markov chains were run for 10 million generations and trees were sampled every 1000th generation. The first 20% of trees, representing the burn-in phase of the analyses, were discarded and the remaining trees were used for calculating the PP value in the majority rule consensus tree.

Phylogenetic trees were viewed using FigTree v1.4.0 [45] and modified in Microsoft Office PowerPoint 2019 and converted to a jpg file using Adobe Photoshop CS6 Extended 10.0 (Adobe Systems, San Jose, CA, USA). The new sequences derived from this study were deposited in GenBank.


**Key to Accepted Montagnula Species**
1. Ascospores are didymosporous 21. Ascospores are phragmosporous 131. Ascospores are dictyosporous 202. Didymospores with sheath32. Didymospores without sheath103. Didymospores surrounded by a mucilaginous sheath43. Sheath was drawn out to form polar appendages84. Ascospores are fusiform
*M. krabiensis*
4. Ascospores are ellipsoidal55. Ascospores are asymmetrical
*M. vakrabeejae*
5. Ascospores are symmetrical66. Asci are (4–)6–8-spored
*M. chromolaenae*
6. Asci are 8-spored7 7. Ascospores brown, slightly constricted at the septum
*M. graminicola*
7. Ascospores dark brown, not constricted at the septum
*M. palmacea*
8. Ascospores 1-seriate, yellowish brown to brown
*M. appendiculata*
8. Ascospores 2–3-seriate99. Ascomata 300–400 × 350–400 μm, asci 84–135 × 10–15 μm
*M. guiyangensis*
9. Ascomata 150–220 × 200–230 μm, asci 60–75 × 8–11 μm
*M. chiangraiensis*
10. Ascomata superficial
*M. longipes*
10. Ascomata immersed or erumpent1111. Ascomata 140–180 × 150–200 µm, not more than 200 µm
*M. acacia*
11. Ascomata greater than 200 μm1212 Ascospores brown, 12–17 × 4–6.5 μm
*M. donacina*
12 Ascospores pale brown, 19–25 × 9–13 μm
*M. opulenta*
13. Ascomata superficial
*M. camporesii*
13. Ascomata immersed or erumpent1414. Asci with short stalks1514. Asci with long pedicellate1615. Ascospores 5 transverse septa, 21–25 × 5–7 μm
*M. subsuperficialis*
15. Ascospores 3 transverse septa, 24–35 × 7.5–14 μm
*M. aquatica*
16. Ascospores with 2 transverse septa
*M. bellevaliae*
16. Ascospores with 3 transverse septa1717. Asci not more than 100 μm
*M. jonesii*
17. Asci greater than 100 μm1818. Ascospores greater than 30 μm, ovoid to ellipsoid
*M. aloes*
18. Ascospores not more than 30 μm, ellipsoid to fusiform1919. Ascomata 385–415 × 510–525 μm, asci 84.5–119.5 × 10.5–13.5 μm
*M. cirsii*
19. Ascomata 300–320 × 300–360 μm, asci 110–130 × 14–20 μm
*M. scabiosae*
20. Sporous transverse septa more than 102120. Sporous transverse septa not more than 102321. Sporous transverse septa more than 15, ascospores 40–45 × 15–17 μm
*M. gigantea*
21. Sporous transverse septa not more than 152222. Ascospores 32–40 × 8–9.8 μm, asci 80–110 × 13–15 μm
*M. dura*
22. Ascospores 31–45 × 13.5–16.5 μm, asci 110–160 × 13–6 μm
*M. triseti*
23. Sporous transverse septa not more than 52423. Sporous transverse septa more than 52724. Ascospores without sheath2524. Ascospores with sheath2625. Ascospores 2–3 transverse septa, 0–1 longitudinal septum, 12.5–16.5 × 4.8–6.5 μm
*M. baatanensis*
25. Ascospores 5 transverse septa, 1 longitudinal septum, 24–29 × 9–11 μm
*M. infernalis*
26. Ascospores 17.5–23 × 5.5–8.5 μm, fusiform to somewhat broadly fusiform
*M. opuntiae*
26. Ascospores 16–18 × 6–7.5 μm, ellipsoid fusoid
*M. thuemeniana*
27. Ascospores without sheath2827. Ascospores with sheath2928. Ascospores 17–25 × 7.5–10 μm, 5–7 transverse septa
*M. obtusa*
28. Ascospores 39–47 × 15–19 μm, 7–9 transverse septa
*M. opaca*
29. Ascospores broadly ellipsoid, 5–7 transverse septa
*M. phragmospora*
29. Ascospores obovoid fusoid, 7(–10) transverse septa3030. Ascospores 2–3 longitudinal septa, 40.8–52 × 17.6–22.4 μm
*M. mohavensis*
30. Ascospores 1–2 longitudinal septa3131. Ascospores 35–50 × 16–20 μm, asci 2–8-spored
*M. dasylirionis*
31. Ascospores 27–42 × 12–15 μm, asci 4–8-spored
*M. yuccigena*



## Figures and Tables

**Figure 1 plants-12-00738-f001:**
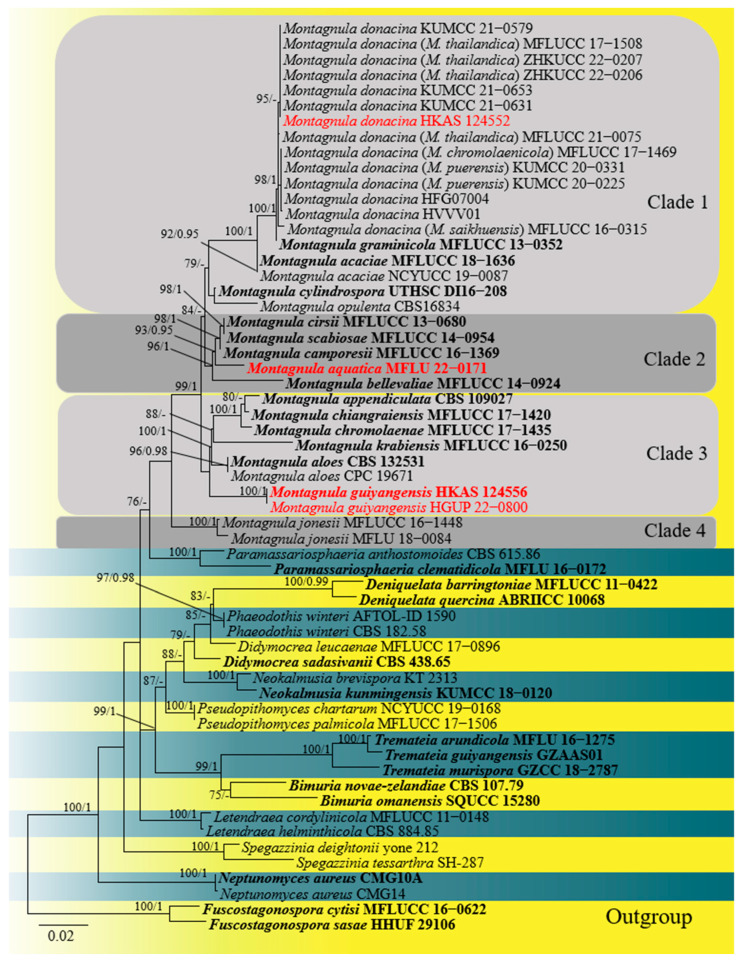
The ML tree based on a combined dataset of LSU, ITS, SSU, and tef1-α sequence data. The tree was rooted with *Fuscostagonospora sasae* (HHUF 29106) and *F. cytisi* (MFLUCC 16–0622). Bootstrap support values for ML greater than 75% and Bayesian posterior probabilities greater than 0.95 are given near the nodes, respectively. Ex-type strains are in bold, the new isolates are in red.

**Figure 2 plants-12-00738-f002:**
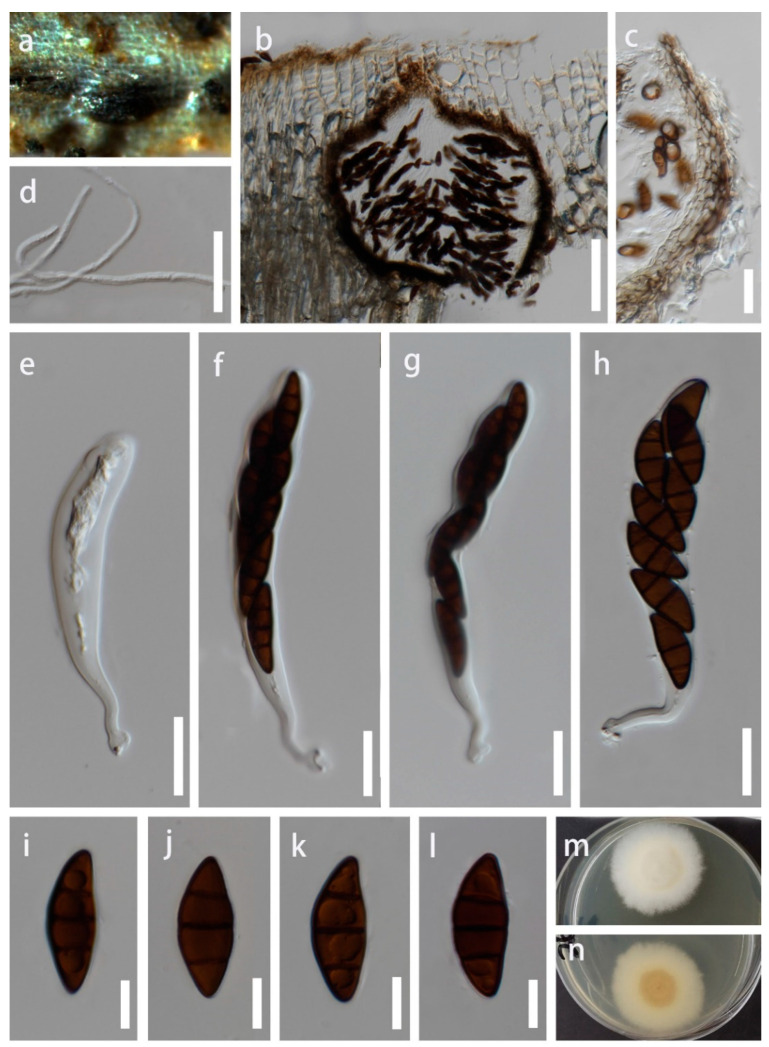
*Montagnula aquatica* (MFLU 22–0171, holotype). (**a**) Appearance of ascomata on the substrate, (**b**) Section through ascomata, (**c**) Peridium, (**d**) Trabeculate pseudoparaphyses, (**e**–**h**) Immature and mature asci, (**i**–**l**) Ascospores, (**m**,**n**) Colony on PDA medium. Scale bars: (**b**) = 100 μm, (**c**–**h**) = 20 μm, (**i**–**l**) = 10 μm.

**Figure 3 plants-12-00738-f003:**
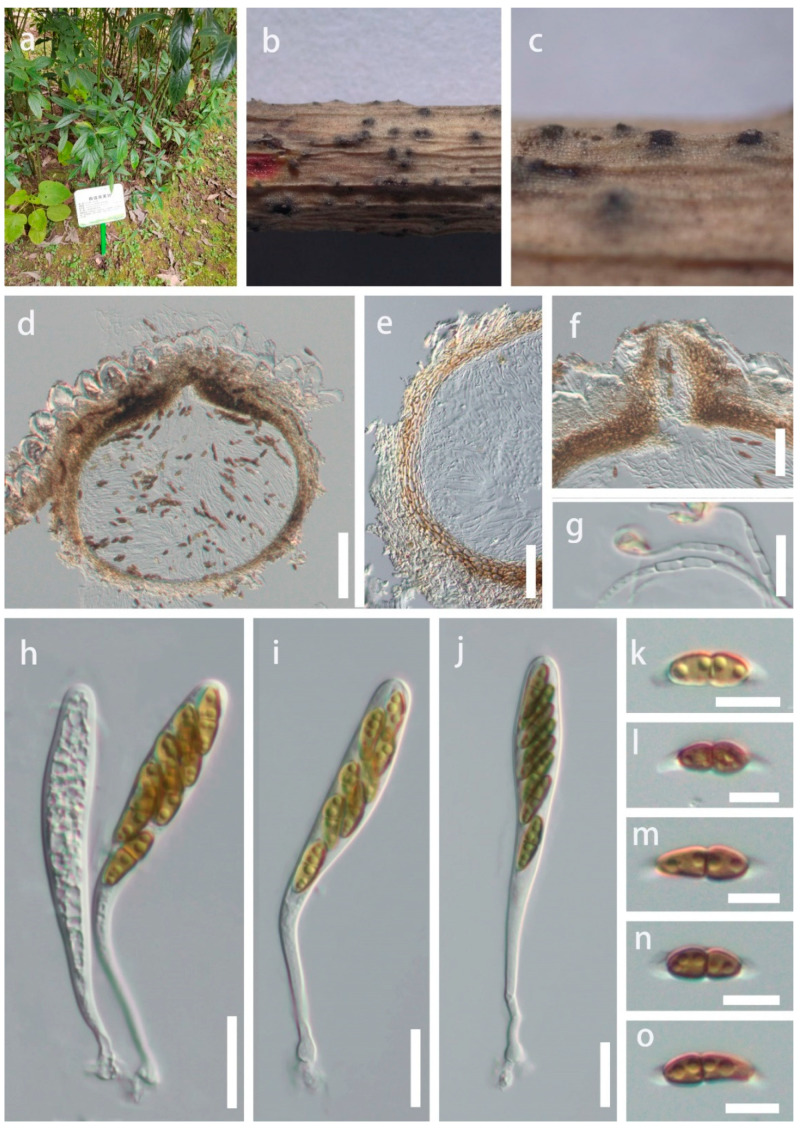
*Montagnula guiyangensis* (HKAS 124556, holotype). (**a**) Host, (**b**,**c**) Appearance of ascomata on the substrate, (**d**) Section through ascomata, (**e**) Peridium, (**g**) Trabeculate pseudoparaphyses, (**h**–**j**) Asci, (**k**–**o**) Ascospores. Scale bars: d = 100 μm, (**e**,**f**) = 50 μm, (**g**–**j**) = 20 μm, (**k**–**o**) = 10 μm.

**Figure 4 plants-12-00738-f004:**
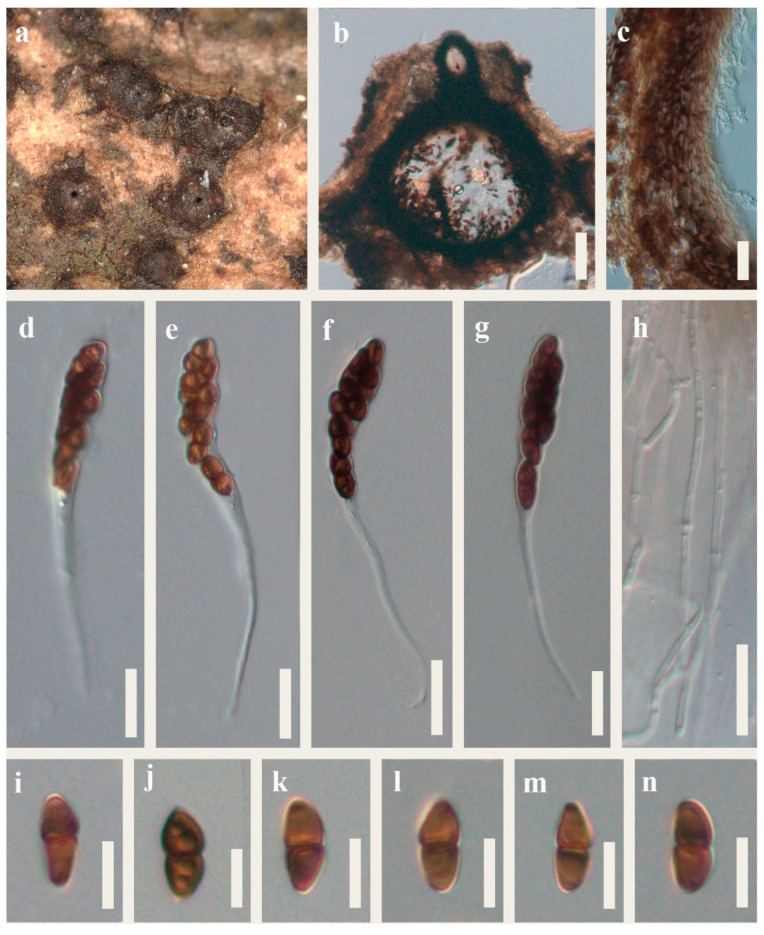
*Montagnula donacina* (HKAS 124552). (**a**) Appearance of ascomata on the substrate, (**b**) Section through ascomata, (**c**) Peridium, (**h**) Paraphyses, (**d**–**j**) Asci, (**i**–**n**) Ascospores. Scale bars: (**b**) = 100 μm, (**c**–**h**) = 20 μm, (**i**–**n**) = 10 μm.

**Table 1 plants-12-00738-t001:** The number of polymorphic nucleotide differences between *M. aquatica* (tef1-*α* not available) and *M. camporesii*, *M. cirsii*, and *M. scabiosae* (without gap).

Species	Strain	ITS (504 bp)	LSU (842 bp)	SSU (971 bp)
*M. camporesii*	MFLUCC 16–1369	14 (2.7%)	7 (0.8%)	4 (0.4%)
*M. cirsii*	MFLUCC 13–0680	14 (2.7%)	7 (0.8%)	2 (0.2%)
*M. scabiosae*	MFLUCC 14–0954	15 (2.9%)	7 (0.8%)	6 (0.6%)

**Table 2 plants-12-00738-t002:** The number of polymorphic nucleotide differences between *M. guiyangensis* HKAS 124556 and *M. aloes*, *M. appendiculata*, *M. chiangraiensis*, and *M. chromolaenae* (without gap).

Species	Strain	ITS (504 bp)	LSU (842 bp)	SSU (971 bp)
*M. aloes*	CBS 132531	20 (4%)	10 (1.2%)	not available
*M. appendiculata*	CBS 10927	18 (3.6%)	16 (1.9%)	not available
*M. chiangraiensis*	MFLUCC 17–1420	13 (2.6%)	18 (2.1%)	27 (2.8%)
*M. chromolaenae*	MFLUCC 17–1435	14 (2.7%)	18 (2.1%)	32 (3.3%)

**Table 3 plants-12-00738-t003:** Morphological comparison of *M. donacina* and *M. chromolaenicola*, *M. puerensis*, *M. saikhuensis*, and *M. thailandica*.

Species	Ascomata	Asci	Ascospores	Reference
*M. chromolaenicola*	Solitary, scattered, semi-immersed to erumpent, brown to dark brown, globose to obpyriform, 300–320 × 215–310 μm, ostiole papilla	Bitunicate, elongate-clavate, 8-spored, 80–100 × 10–13 μm, long pedicel	Broadly fusiform to ellipsoid, brown to dark brown, overlapping 1–2-seriate, 1 transverse septum, constricted at the septum, asymmetrical, 15–17 × 5–6.5 μm	Mapook et al. [8]
*M. donacina*	Ascostromata with flat bottom, gregarious, immersed to erumpent, black, globose to pyriform, 500 μm, pseudothecial ostiole	Bitunicate, clavate, 8-spored, with a long pedicel	Ellipsoid, brown, 1–2-seriate, 1 transverse septum, constricted at the septum, asymmetrical, 12–17 × 4–6.5 μm	Aptroot [13], Pitt et al. [28]
*M. puerensis*	Solitary, scattered, semi-immersed, black, globose, 300–600 × 230–380 μm, with a central ostiole	Bitunicate, elongate-clavate, 8-spored, 70–105 × 10–15 μm, long pedicel	Ellipsoid, brown to dark brown, biseriate, 1 transverse septum, constricted at the septum, asymmetrical, 10–20 × 4–7 μm	Du et al. [21]
*M. saikhuensis*	Solitary, scattered, immersed, globose, brown to dark brown, 400–450 × 400–500 μm, ostiolate	Bitunicate, elongate-clavate to short-cylindrical, 8-spored, 70–100 × 10–12μm, long pedicel	Ellipsoid, brown to blackish brown, overlapping 1–2-seriate, 1 transverse septum, unequally and strongly constricted at the septum, asymmetrical, 12–16 × 4–6 µm	Wanasinghe et al. [14]
*M. thailandica*	Solitary, scattered, immersed to erumpent, brown to dark brown, globose to obpyriform, brown to dark brown, 405–415 × 330–350 μm, ostiole papillate	Bitunicate, elongate-clavate, slightly curved, 8-spored, 80–100 × 9–15 μm, long pedicel	Broadly fusiform to ellipsoid, brown to reddish-brown, overlapping 1–2-seriate, 1 transverse septum, constricted at the septum, asymmetrical, 14–17 × 4.5–7.5 μm	Mapook et al. [8]

**Table 4 plants-12-00738-t004:** Comparison of nucleotide differences between *M. donacina* (KUMCC 21–0653) and *M. chromolaenicola*, *M. puerensis*, *M. saikhuensis*, and *M. thailandica*.

Species	Strain	ITS (504 bp)	LSU (842 bp)	SSU (971 bp)	tef1-α (840 bp)
*M. chromolaenicola*	MFLUCC 17–1469	0	3	1	10
*M. puerensis*	KUMCC 20–0225	0	0	1	8
*M. saikhuensis*	MFLUCC 16–0315	4	2	3	not available
*M. thailandica*	MFLUCC 17–1508	3	2	3	2

**Table 5 plants-12-00738-t005:** Primers and PCR procedures used in this study.

Locus	Primers		PCR Procedures	Reference
	Name	Sequence (5′-3′)		
LSU	LR0R	ACCCGCTGAACTTAAGC	94 °C—3 min; 94 °C—30 s; 52 °C—30 s; 72 °C—1 min; Repeat 2–4 for 35 cycles; 72 °C—8 min; 4 °C on hold	White et al. [34], Rehner and Samuels [35]
	LR5	TCCTGAGGGAAACTTCG		
SSU	NS1	GTAGTCATATGCTTGTCTC		White et al. [34]
	NS4	CTTCCGTCAATTCCTTTAAG		
ITS	ITS5	GGAAGTAAAAGTCGTAACAAGG		
	ITS4	TCCTCCGCTTATTGATATGC		
tef1-α	EF1-983F	GCYCCYGGHCAYCGTGAYTTYAT	94 °C 2 min; 36 cycles of 66 °C–56 °C (touchdown 9 cycles), 94 °C 30 s, 56 °C 1 min, 72°C 1 min; 72 °C 10 min; 4 °C on hold	Rehner and Buckley [36]
	EF1-2218R	ATGACACCRACRGCRACRGTYTG		

**Table 6 plants-12-00738-t006:** Sequence data were used for phylogenetic analyses with the corresponding GenBank accession numbers. The newly generated strains are in red. N/A: Not available.

Species	Strain/Voucher No.	LSU	SSU	ITS	*tef-α*
*Bimuria omanensis*	SQUCC 15280	NG_071257	N/A	NR_173301	MT279046
*Bimuria novae-zelandiae*	CBS 107.79	AY016356	N/A	MH861181	DQ471087
*Deniquelata barringtoniae*	MFLUCC 11−0422	JX254655	JX254656	NR_111779	N/A
*Deniquelata quercina*	ABRIICC 10068	MH316157	MH316155	MH316153	N/A
*Didymocrea leucaenae*	MFLUCC 17−0896	NG_066304	MK347826	NR_164298	MK360052
*Didymocrea sadasivanii*	CBS 438.65	DQ384103	DQ384066	MH870299	N/A
*Fuscostagonospora sasae*	HHUF 29106	NG_059395	NG_061003	NR_153964	AB808524
*Fuscostagonospora cytisi*	MFLUCC 16−0622	KY770978	KY770977	N/A	KY770979
*Letendraea cordylinicola*	MFLUCC 11−0148	NG_059530	NG_068362	NR_154118	N/A
*Letendraea helminthicola*	CBS 884.85	AY016362	AY016345	MK404145	MK404174
*Montagnula acaciae*	MFLUCC 18−1636	ON117298	ON117267	ON117280	ON158093
*Montagnula acaciae*	NCYUCC 19−0087	ON117299	ON117268	ON117281	ON158094
*Montagnula aloes*	CPC 19671	JX069847	N/A	JX069863	N/A
*Montagnula aloes*	CBS 132531	NG_042676	N/A	NR_111757	N/A
* Montagnula aquatica *	MFLU 22−0171	OP605986	OP600504	OP605992	N/A
*Montagnula appendiculata*	CBS 109027	AY772016	N/A	DQ435529	N/A
*Montagnula bellevaliae*	MFLUCC 14−0924	KT443902	KT443904	KT443906	KX949743
*Montagnula camporesii*	MFLUCC 16−1369	NG_070946	NG_068418	MN401746	MN397908
*Montagnula chiangraiensis*	MFLUCC 17−1420	NG_068707	NG_070155	NR_168864	N/A
*Montagnula chromolaenae*	MFLUCC 17−1435	NG_068708	NG_070156	NR_168865	N/A
*Montagnula cirsii*	MFLUCC 13−0680	KX274249	KX274255	KX274242	KX284707
*Montagnula cylindrospora*	UTHSC DI16-208	LN907351	N/A	LT796834	LT797074
*Montagnula donacina*	HVVV01	KJ628377	KJ628376	KJ628375	N/A
*Montagnula donacina*	HFG07004	MF183940	N/A	MF967419	N/A
*Montagnula donacina*	KUMCC 21−0653	OP059052	OP059003	OP058961	OP135938
*Montagnula donacina*	KUMCC 21−0579	OP059054	OP059005	OP058963	OP135940
*Montagnula donacina*	KUMCC 21−0631	OP059053	OP059004	OP058962	OP135939
* Montagnula donacina *	HKAS 124552	OP605987	N/A	OP605991	N/A
*Montagnula donacina (M. chromolaenicola)*	MFLUCC 17−1469	NG_070948	NG_070157	NR_168866	MT235773
*Montagnula donacina (M. puerensis)*	KUMCC 20−0225	MW575866	MW575864	MW567739	MW573959
*Montagnula donacina (M. puerensis)*	KUMCC 20−0331	MW575867	MW575865	MW567740	MW573960
*Montagnula donacina (M. saikhuensis)*	MFLUCC 16−0315	KU743210	KU743211	KU743209	N/A
*Montagnula donacina (M. thailandica)*	MFLUCC 17−1508	NG_070949	NG_070158	MT214352	MT235774
*Montagnula donacina (M. thailandica)*	MFLUCC 21−0075	MZ538549	N/A	MZ538515	N/A
*Montagnula donacina (M. thailandica)*	ZHKUCC 22−0206	OP297777	OP297791	OP297807	OP321576
*Montagnula donacina (M. thailandica)*	ZHKUCC 22−0207	OP297778	OP297792	OP297808	OP321577
*Montagnula graminicola*	MFLUCC 13−0352	KM658315	KM658316	KM658314	N/A
* Montagnula guiyangensis *	HKAS 124556	OP600484	OP600500	OP605989	N/A
* Montagnula guiyangensis *	HGUP 22−800	OP600485	OP600501	OP605990	N/A
*Montagnula jonesii*	MFLUCC 16−1448	KY273276	KY313618	KY313619	KY313620
*Montagnula jonesii*	MFLU 18−0084	ON117300	ON117269	ON117282	ON158095
*Montagnula krabiensis*	MFLUCC 16−0250	NG_068826	NG_068385	NR_168179	MH412776
*Montagnula opulenta*	CBS 16834	NG_027581	AF164370	AF383966	LT797074
*Montagnula scabiosae*	MFLUCC 14−0954	KT443903	KT443905	KT443907	N/A
*Neokalmusia brevispora*	KT 2313	AB524601	AB524460	NR_154262	AB539113
*Neokalmusia kunmingensis*	KUMCC 18−0120	MK079889	MK079887	MK079886	MK070172
*Neptunomyces aureus*	CMG10A	N/A	N/A	MK912119	MK947998
*Neptunomyces aureus*	CMG14	N/A	N/A	MK912123	MK948002
*Paramassariosphaeria anthostomoides*	CBS 615.86	MH873693	GU205246	MH862005	N/A
*Paramassariosphaeria clematidicola*	MFLU 16−0172	KU743207	KU743208	KU743206	N/A
*Phaeodothis winteri*	AFTOL-ID 1590	DQ678073	DQ678021	N/A	DQ677917
*Phaeodothis winteri*	CBS 182.58	GU301857	GU296183	N/A	N/A
*Pseudopithomyces chartarum*	NCYUCC 19−0168	MW063220	MW079349	MW063159	N/A
*Pseudopithomyces palmicola*	MFLUCC 17−1506	MT214447	N/A	MT214353	N/A
*Spegazzinia deightonii*	yone 212	AB807582	AB797292	N/A	AB808558
*Spegazzinia tessarthra*	SH 287	AB807584	N/A	JQ673429	AB808560
*Tremateia arundicola*	MFLU 16−1275	KX274248	KX274254	KX274241	KX284706
*Tremateia guiyangensis*	GZAAS01	KX274247	KX274253	KX274240	KX284705
*Tremateia murispora*	GZCC 18−2787	MK972751	MK972750	NR_165916	MK986482

## Data Availability

Not applicable.

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
