# Peer review of "Morphology and Phylogeny Reveal Three Montagnula Species from China and Thailand"

_plants, 2023, doi:10.3390/plants12040738_

Round 1

Reviewer 1 Report

The paper entitled "" can be publish. However, the authors need to explain in more detail why the work is scientifically important in their Introduction. As well as the importance of the main findings in the Discussion.

1. The authors need to clarify "Four fresh collections" Line 13, as well as "Several similar species' Line 18.

2. The authors need to include why Montagnula is important in the Introduction.

3. The authors need to explain "from terrestrial and fresh water habitats in China and Thailand" Line 47 in the Introduction. For example, what do they expect to find when sampling in the different environments (countries and habits). Also is this information need to be address in the remainder of the paper.

4. The authors also need to provide all of the necessary sample information in the Materials and Methods.

Author Response

Dear Reviewer,

Thanks for your comments. We have carefully considered the comments and tried our best to address every one of them. We hope the manuscript after careful revisions meet your high standards comments.

Below we provide the point-by-point responses. All modifications in the revised manuscript have been used the “Track Changes”.

Reviewer 1:

  1. The authors need to clarify "Four fresh collections" Line 13, as well as "Several similar species' Line 18.

R: We changed this sentence to “Four strains were isolated from two fresh twigs of Helwingia himalaica and two dead woods during investigations of micro-fungi in China and Thailand”. We changed “Several similar species” to “Montagnula chromolaenicola, M. puerensis, M. saikhuensis and M. thailandica are discussed and synonymized under M. donacina”.

  1. The authors need to include why Montagnula is important in the Introduction.

R: Thanks for your suggestion. We supplemented the importance of Montagnula in the Introduction section as follows:

So far, there are 39 validly published Montagnula species in Species Fungorum (accessed on 10 Jan 2023) [15]. However, only 18 species have molecular data. Morphologically, sexual morphs of Montagnula have three different ascospores (didymospore, phragmospore and dictyospore) [8,16]. Phylogenetically, species with the same type of ascospore tend to cluster together [9,17]. In recent years, there have been many reports on Montagnula species [8,9,18-20], but there are very few comprehensive and systematic paper.

  1. The authors need to explain "from terrestrial and fresh water habitats in China and Thailand" Line 47 in the Introduction. For example, what do they expect to find when sampling in the different environments (countries and habits). Also is this information need to be address in the remainder of the paper.

R: Thanks for your suggestion. We corrected this sentence to “To study the taxonomy and diversity of Montagnula species, four Montagnula specimens were obtained from terrestrial and freshwater habitats in China and Thailand.”

  1. The authors also need to provide all of the necessary sample information in the Materials and Methods.

R: Thanks for your comments. We provided the information for samples in the Taxonomy part. It will be repeated if more information are provided in the Materials and methods.

Reviewer 2 Report

In this manuscript (ID: plants-2160042), the author's effort was to study the phylogenetically identification and taxonomic assignment for Montagnula species from China and Thailand, which has a significance to the scientific community.

The authors, however, should consider the results in context of findings by previous studies on the Montagnula species. It will be easier for the readers to follow the entire story. For instance, the new species Montagnula acaciae has been described through recent investigations. Additionally, authors should consider adding all the publicly-available DNA sequences as background information in the revised manuscript.

The authors should explain to the readers why they used the large subunit (28S, LSU), small subunit (18S, SSU), and internal transcribed spacers (ITS1- 5.8S-ITS2). Are LSU, SSU and ITS the best biomarkers for fungal identification at the species level? Are LSU, SSU and ITS sufficient for fungal species delineating when there are closely-related species ? Are there recent studies that used the translation elongation factor 1-alpha (tef1), partial RNA polymerase II subunit (rpb2) and b-tubulin (tub2) genes for the molecular identification of Montagnula species? The authors should consider adequately describing their results in context of findings by previous studies. It should be also for the Discussion section.

I hope it helps.

Author Response

Dear reviewer,

Thanks for your comments. We have carefully considered the comments and tried our best to address every one of them.  We hope the manuscript after careful revisions meet your high standards comments.

Below we provide the point-by-point responses. All modifications in the revised manuscript have been used the “Track Changes”.

Reviewer 2:

  1. The authors, however, should consider the results in context of findings by previous studies on the Montagnula It will be easier for the readers to follow the entire story. For instance, the new species Montagnula acaciae has been described through recent investigations. Additionally, authors should consider adding all the publicly-available DNA sequences as background information in the revised manuscript.

R: Thank you for your suggestions. We referred to previous studies and summarize them. The results are presented in the Introduction, as follows:

Berlese [10] introduced Montagnula, typified by M. infernalis, which has bitunicate asci and dictyosporous ascospores. About a century later, Crivelli [11] refined Pleospora and transferred eight Pleospora species and one Teichospora species to Montagnula based on morphology. Leuchtmann [12] included phragmosporous and didymosporous species in this genus, making species identification heterogeneous. Aptroot [13] established Munkovalsaria to accommodate Mu. donacina based on valsoid ascomata, bitunicate, fissitunicate asci and 1-septate ascospores but Wanasinghe et al. [14] synonymized Munkovalsaria under Montagnula based on analyses of combined LSU, SSU and ITS sequence data. Crous et al. [7] reported the first coelomycetous asexual morph species M. cylindrospora in this genus.

In addition, we have added the recently published sequences of Montagnula species in the revised manuscript. All available Montagnula sequences are presented in the phylogenetic analysis.

  1. The authors should explain to the readers why they used the large subunit (28S, LSU), small subunit (18S, SSU), and internal transcribed spacers (ITS1- 5.8S-ITS2). Are LSU, SSU and ITS the best biomarkers for fungal identification at the species level? Are LSU, SSU and ITS sufficient for fungal species delineating when there are closely-related species? Are there recent studies that used the translation elongation factor 1-alpha (tef1), partial RNA polymerase II subunit (rpb2) and b-tubulin (tub2) genes for the molecular identification of Montagnula species? The authors should consider adequately describing their results in context of findings by previous studies. It should be also for the Discussion section.

R: Thanks for your suggestions. We selected LSU, SSU and ITS genes based on previous studies. tef1gene has been used recently for Montagnula and we added it in our revised manuscript. rpb2 and tub2 genes are not available for any Montagnula species. In the Discussion section, we discussed different clades which were also corresponding to previous studies.

Round 2

Reviewer 2 Report

From the review perspective, I invite authors to consider the results in context of findings by recent studies on the new species Montagnula acaciae that has been described by Tennakoon et al., 2022. Authors should consider adding the publicly-available DNA sequences from M.  acaciae in the revised manuscript. It could also improve the discussion of the manuscript.

Reference

Tennakoon DS, Thambugala KM, de Silva NI, Suwannarach N, Lumyong S. A taxonomic assessment of novel and remarkable fungal species in Didymosphaeriaceae (Pleosporales, Dothideomycetes) from plant litter. Front Microbiol. 2022; 13:1016285. doi: 10.3389/fmicb.2022.1016285.

Author Response

Dear Reviewer,

Thanks for your comments. We added the DNA sequences from M.  acaciae in the phylogenetic analysis and the result is presented in the revised manuscript. We also added this species in the key to Montagnula species.

Thank you very much.

Sincerely,

Ya-Ru Sun

Jing-Yi Zhang

Kevin D. Hyde

Yong Wang

Ruvishika S. Jayawardena